# The Attitudinal Style as a Pedagogical Model in Physical Education: Analysis of Its Effects on Initial Teacher Training

**DOI:** 10.3390/ijerph17082816

**Published:** 2020-04-19

**Authors:** Ángel Pérez-Pueyo, David Hortigüela-Alcalá, Alejandra Hernando-Garijo, Antonio Granero-Gallegos

**Affiliations:** 1Physical and Sports Education Area, Faculty of Physical Activity and Sport Sciences, University of León, 24007 León, Spain; angel.perez.pueyo@unileon.es; 2Department of Specific Didactics, Faculty of Education, University of Burgos, 09001 Burgos, Spain; dhortiguela@ubu.es (D.H.-A.); ahgarijo@ubu.es (A.H.-G.); 3Department of Education, Faculty of Education Sciences, Health Research Center, University of Almeria, 04120 Almeria, Spain

**Keywords:** Attitudinal Style, physical education, initial teacher training, teaching role, transferability of learning

## Abstract

The implementation of the pedagogical model has meant an increase in rigour and coherence in Physical Education (PE) classes. The objectives of the study were twofold; (a) to delimit the characteristics and elements that make up Attitudinal Style as a pedagogical model; (b) to analyze the perception of future teachers on the usefulness and transferability of the model in their classes. Twelve future PE teachers (seven women and five men) with an age of 20.14 ± 1.48 participated. All of them were part of the University of Burgos (Spain). A qualitative approach was used with two data collection instruments (reflective group journals and discussion group) and two categories of analysis were established: (a) usefulness in the construction of professional identity; (b) transferability of the Attitudinal Style in the school. The results showed how future PE teachers consider the model as a transcendental methodological tool in understanding and addressing PE at school. Interpersonal relationships in the classroom, student autonomy and group responsibility are highlighted as necessary aspects with high transferability to the school.

## 1. Introduction

The potential of physical education (PE) is unquestionable in encouraging student motivation towards learning and adherence to physical activity [1,2]. To this end, it is necessary to provide future teachers with tools to make the teaching and learning processes useful, meaningful, coherent and replicable. Otherwise, feelings of frustration will take hold of the students and generate demotivation towards the subject [3]. As established by Hortigüela, Pérez-Pueyo, and Fernández-Río [4], the social and cultural models of the countries directly affect the way in which PE is taught in initial teacher training, having repercussions on their way of understanding the teaching of PE.

In this sense, it is important to emphasize that the quality of future teachers will depend largely on the training they receive, which gives great importance to initial teacher training. Da Costa, McNamee, and Lacerda [5] state that the quality of training depends on variables such as the typology of the faculty, teaching methodology, skills to be developed and the design of the objectives of the subject. In fact, as commented by Pill, Penney, and Swabey [6], the subject of PE’s pedagogy and sport in universities is usually approached from very different angles, which means that the learning objectives are not always clear [7]. However, it seems fundamental that students acquire different teaching approaches in the subjects linked to sport pedagogy to encourage social inclusion of the individual through the practice of physical activity [8]. Such knowledge and learning will allow future teachers to choose the most appropriate approach for their students. Under this approach of educational coherence, Karp, Scruggs, Brown, and Kelder [9] indicate that it is only possible to improve the results in the initial training of the teaching staff if teaching processes of PE are carried out under common, coherent criteria, have direct application to the classroom, and are adaptable to different contexts (recreation, free time, etc.). This would increase student motivation and connect the university and the school [10]. Changing from teacher-centred teaching to student-centred teaching [11] generates in the students an increase in motivation, responsibility, and autonomy towards work.

Within the Spanish context, in recent years a series of publications have been established that specify the most recognized pedagogical models organized into basic and emerging [12,13], and have analyzed how they have evolved from teaching styles [14,15,16] to pedagogical models that have been agreed upon as those that should be taught [17]. Initial teacher training in PE becomes a key element in building the professional identity of future teachers throughout their lives. In this sense, the experimentation of these pedagogical models will allow them to generate evidence of its operation and applicability in the classroom, a key aspect in their training. There are many models, although among the so-called emerging ones and those with the highest level of dissemination and production is the Attitudinal Style.

Therefore, the present article addresses two objectives: (a) to delimit the characteristics and elements that make up Attitudinal Style as a pedagogical model; (b) to analyse the perception of future teachers on the usefulness and transferability of the model in their classes.

### Teaching Pedagogical Models in the Initial Training of Physical Education Teachers: The Attitudinal Style

Cañadas, Santos-Pastor, and Castejón [18] establish that in order for teachers to achieve adequate mastery of what and how to teach, it is necessary to be able to design tasks with which students can obtain real learning. In order to do so, future teachers must know and put into practice teaching techniques, styles, and strategies that make it possible to evaluate student competence [19,20].

When we go into PE, it has been proven that mastering the content that is intended to be taught and, above all, how to teach it generates a significant improvement in student learning [21,22,23,24,25,26]. In this sense, although in the last three decades the teaching of teaching styles [14,15] and their contextualization for Spain [16,27] have been the fundamental reference on how to teach [17], at present, this reference must be fundamentally linked to pedagogical models [12,13]. In this sense, there are two reasons for tackling this specific training from the knowledge of the pedagogical models: (a) firstly, because research in relation to the models has demonstrated the suitability of their application with respect to the involvement of the students and the learning generated; (b) secondly, because of the disparate perception that the traditional teaching approaches have generated among students, graduates, and teachers.

Several studies have analyzed the interests of using student-centered teaching styles and methodologies [28,29,30]. Zapatero’s research [31] establishes that the use of student-centered methodologies and styles is less frequent than the use of methods that favour teacher control [32,33,34,35,36]. This has shown that after three decades of the use of teaching styles in Spain [27,32,37,38,39,40], they seem to have had little influence on adequate teacher practice. In fact, reviews of student-centred methodologies state that PE teachers [28,41,42] recognise the need to vary the methodological approach towards those that favour student involvement and motivation. However, despite the belief in the need for this methodological change, its impact is still not high in the classroom [31,33,35,43,44]. 

The Attitudinal Style began its journey in León (Spain) at the end of the 90s, culminating with the completion of a doctoral thesis in 2004 [45]. Considered to be one of the emerging pedagogical models [12,13], it establishes attitudes as the backbone for better learning and greater motivation towards PE. Furthermore, it proposes the use of the motor as a means (and not as an end), working simultaneously and in balance with the rest of the capacities that develop the individual in an integral manner (cognitive–intellectual, affective–motivational, interpersonal relations, and social insertion) [46,47]. Its purpose is that all students have positive experiences without exception and with inclusion [48], generating a real group that cooperates or collaborates. There are three components through which it is developed: (1) Intentional Corporal Activities, (2) Sequential Organization Towards Attitudes and (3) Final Assemblies [45,49,50,51]. The evolution followed by this pedagogical model, linked to the creation of the Interdisciplinary and Interdisciplinary Working Group Attitudes in 2007 (www.grupoactitudes.com) and the interaction networks established with different groups, has generated a series of reorientations and incorporations that have undoubtedly benefited the proposal. Among these, two stand out: (a) the proposal of competences through the so-called INCOBA Project (Project for the Integration of Basic Competences) [52,53,54,55,56], and (b) the incorporation of formative evaluation associated with self-evaluation and co-evaluation processes [56,57] through the design [58] and elaboration of new evaluation and qualification instruments [59,60].

The Attitudinal Style is a pedagogical model with a global character since it can be implemented in any curriculum content. It focuses on the learning process and the needs of the students. In fact, its applicability to any content allows its hybridization with other models [52,56]. One of the aspects that characterizes it is the innumerable amount of pedagogical and didactic material generated in these 25 years, partly collected in the bibliographical review of Tena [61] and which can be downloaded free of charge from the web (www.grupoactitudes.com) or from repositories such as https://www.researchgate.net/. Examples of its pedagogical and didactic application can be found in physical conditions [45], intentional games without elimination [56], sports such as football [62], basketball [57] or Gaelic football [63], opposing sports [64], drama [65], acrobatics [66], dance [67], shadow theatre [68], activities in the natural environment such as knots and obstacle breaking [69], and street work [51]. 

Therefore, the main aspects that characterize it are: (a) that the guidelines in its teaching are very clear and can be analyzed in detail in all the didactic and pedagogical publications generated in the last decades; (b) that it can be applied in diverse contexts and in any type of content; (c) that it guarantees its replicability by other teachers in diverse contexts. In relation to the latter, the research carried out at the end of the 1990s and which led to the author’s doctoral thesis [45] showed that the results obtained through the Attitude Scale for Integrated Physical Education (EAEFI) strongly expressed the improvement of the attitude towards the PE of students receiving the Attitudinal Style.

After years of pedagogical and didactic production, research continued and Hortigüela, Fernández-Río, and Pérez-Pueyo [70] evaluated the effects of the prolonged use of a traditional teaching approach and the Attitudinal Style by finding that students who experienced the Attitudinal Style perceived the PE class to be significantly more useful than with the traditional approach. When the two approaches were compared in teaching football [71], the groups that experienced the Attitudinal Style developed a more task-oriented perception of the classroom climate than those who received the traditional approach. In relation to the factors implicit in physical self-concept, after having received a physical fitness teaching unit its effectiveness was demonstrated with respect to the positive influence on girls and its direct influence on their self-concept [72]. 

In relation to the responsibility in the assessment [73], they showed that the students who received the Attitudinal Style increased individual and group responsibility in the regulation of work during the process and the authenticity of the acquired learning linked to real life [74]. In the same vein, Hortigüela, Pérez-Pueyo, and Fernández-Río [75] demonstrated the increased level of responsibility of PE students in the assessment process.

## 2. Materials and Methods

### 2.1. Participants

Twelve future PE teachers (7 women and 5 men) with an age of 20.14 ± 1.48 participated. All of them were studying for their Primary Education Degree at the Faculty of Education of the University of Burgos (Spain). As this is a qualitative methodology, the participants were not sought to be representative of the whole. They were intentionally selected according to the criteria of voluntariness, motivation and high academic record in the degree. Specifically, they studied the subject of Physical Education and its didactics, an obligatory subject in the training of future PE teachers. The teacher of the subject was 34 years old; he was a doctor and a specialist in the implementation of pedagogical models, especially in the Attitudinal Style. This professor is one of the researchers of the study, which allowed for granting a validity to the applied design and a greater knowledge of the obtained results.

### 2.2. Instruments

Two different instruments were used for the collection of information. The questions that make up each of the instruments used were structured on the basis of the two categories of the study and to obtain greater specificity of the data [76]. Therefore, these instruments are based on construct validity as they are built specifically in relation to the objectives of the study.

Reflective group diaries: The main objective of the elaboration of the diaries was to reflect on the learning generated throughout the development of the subject. This diary was prepared in groups of four, and all the members of the group discussed the most relevant aspects. It is completed weekly in order to give a periodicity to the experiences of the subject. The teacher throughout the course reviewed this diary to provide constant feedback for its proper implementation. It had a semi-structured character, starting from three categories on which students had to add the information (Table 1). This allowed for two things: on the one hand, uniformity in the collection of data from all the class journals, and on the other, the freedom of each group to expand the reflective and personal information as much as they considered [77].

Discussion groups: They were developed at the end of the process with 12 students who had a high grade in the course. The qualification obtained in the degree and not in the specific subject was taken into account in order to not bias the results obtained. A semi-structured script was used to collect the information (Table 2). All the participants spoke in a proportionate manner about each of the issues raised. The teacher gave the floor in order to encourage dialogue, discussion, and exchange of ideas.

### 2.3. Design and Procedure

The research responds to a retrospective design of a phenomenological nature based on the understanding of educational phenomena from the analysis of the participants’ discourses [78].

The research has been structured in four distinct phases, from February to June 2019:

Phase 1. Structuring of the study and planning of the subject: The theoretical and practical classes were designed throughout the semester. The 24 theoretical classes revolved around the teaching pedagogy of the PE, focusing on knowledge of the curriculum and its adaptation to the classroom. Specifically, aspects related to methodology and evaluation were worked on in depth. In the 32 practical classes, we worked on a diversity of contents: cooperative challenges, acrobatics, judo, juggling, jumping to the combat, collective sports, alternative sports, etc. All these contents were taught through the Attitudinal Style, respecting the phases and elements that characterize it.

Phase 2. Elaboration and revision of the reflexive group diaries: The group work throughout the course elaborated these diaries. With a weekly cadence, the students collected their reflections. The teacher of the subject gave periodic feedback so that the instruments had an adequate quality and collected evaluations of the whole subject. 

Phase 3. Elaboration of the discussion group: After finishing the subject, the discussion group was carried out with the students. It lasted 90 min, and from the beginning the group was told about the importance of their answers for the research. The session was recorded on video for a better recapitulation of the data after viewing. The anonymity of their answers was guaranteed. We sought to deepen the theme of the study in order to reach a reflective conversation in a relaxed atmosphere. 

Phase 4. Analysis of the data by the researchers: The data from the reflective journals and the discussion group were transcribed and placed into the text analysis software Weft QDA. In addition, there was an in-depth reflection on the aims of the study, the procedure carried out, and its suitability for the objectives set.

To start the research, first, permission was obtained from the Ethics Committee of the University of the principal researcher. To this end, the protocol established at the university was followed (https://www.ubu.es/vicerrectorado-de-investigacion-y-transferencia-del-conocimiento/comision-de-bioetica). The students were clearly informed about the purposes of the research. They were encouraged to answer the questions as truthfully as possible and were assured that their answers would not affect their course grades. A formative and shared assessment was used throughout the course, which guarantees the involvement and responsibility of the students in carrying out the tasks. This implies both a constant feedback between teacher and students, and the use of the grading instruments throughout the process. This process of transparency favors the self-regulation of student tasks and the need for the veracity of their answers to justify the work done. 

### 2.4. Data Analysis

A qualitative approach was employed to gain an in-depth understanding of future PE teachers’ perceptions of Attitudinal Style. For this purpose, their experiences and reflections were studied in depth, analyzing the transferability and usefulness of this pedagogical model. The source for obtaining data was the assessments and experiences of those involved in the process. This allows us to reflect on the study phenomenon and how the interactions between the participants influence the very purposes of intervention, focusing mainly on interpretative models [79]. A triangulation was carried out between the information obtained in the data collection instruments, which was very positive as it allowed for a multidimensional analysis [80]. This triangulation was carried out among the data collection instruments used, provided that the information contributed significantly to the study categories. From this triangulation, the most significant text extracts were selected.

In order to guarantee the reliability, transferability, and credibility of the results, the most significant text extracts were coded in each of the instruments using the cross-matching patterns [81]. The researchers took an active part in the field work, reflecting throughout the process on the influence of events.

### 2.5. Generation of Categories and Their Categorization 

Once the data from each instrument used was transcribed, it was placed into the Weft QDA computer and analysis program. Through the saturation of texts and coinciding ideas, the information was grouped into the two categories of the study: (a) utility in the construction of professional identity; (b) transferability of Attitudinal Style in the school. These categories are in relation to the objectives of the study, thus respecting the criteria of specificity and coherence that all qualitative research should have [82].
Usefulness in the construction of professional identity: aspects related to the way in which having received the methodology of the Attitudinal Style has served them to position themselves towards the approach that they want to give to the PE, what their educational goals should be, and what factors directly influence their school treatment.Transferability of the Attitudinal Style in the school: This links all the information related to the way in which the Attitudinal Style can be applied in the school: contents, tasks, student motivation, organization of spaces, materials, and generation of a positive climate in the classroom.


### 2.6. Coding of Data Collection Instruments

Different acronyms are used to identify the text extracts with the data collection instruments from which they came from. In relation to group reflective journals, (DRG) is used. With regard to discussion groups, (GDE) is used. 

## 3. Results

The results are structured according to the categories generated. These categories have been constructed from the different data collection instruments and the objectives of the study. In each category, the most significant text extracts obtained are presented.

### 3.1. Usefulness in the Construction of Professional Identity (242 Text Extracts)

We see how future PE teachers’ value Attitudinal Style as a model has positioned them to approach PE in certain ways:
*“It is usual in physical education to be told about games and diversity of activities, but the methodology used in this subject is allowing us to be aware of how important it is to control all the variables in the classroom […]. “It is very different from the physical education we have experienced as students, since in class we see how important it is to justify the teaching units we do according to the type of students we will have in the classroom, and depending on these adaptations we will achieve our objectives better”* (DRG).
*“You realize throughout the course how physical education is the most relevant subject in the curriculum, since the treatment of the body is very much linked to children’s fears, insecurities and learning. “The Attitudinal Style methodology has helped me to know what kind of PE teacher I want to be and how I will show my students tomorrow.” “Now I always think more than once about each activity, seeing how to present it in the most motivating way possible to the students so that they can get it and learn from it”* (GDE).

The results show how PE’s future teachers fundamentally emphasize the Attitudinal Style in the pursuit of group success in the classroom and the empowerment of students’ abilities:
*“I was afraid at the beginning of the course about what I might find. In many cases at school, I felt excluded by the group […]. In this subject, I would highlight the opposite, as we have always worked as a team and managed to get things done together.” “Sometimes I thought I was incapable when David told me what we were going to do, but the methodology and the way the activities were carried out showed that we always ended up getting them.” “We’re seeing how the easiest thing is to make an excuse not to do an activity… However, we are learning how important it is for all students without exception to be able to achieve, as this strengthens their self-esteem, and here the teachers are key […].”* (DRG).
*“I admit that at the beginning of the course I was quite competitive and individual […]. Now I have realized that solidarity in the group is fundamental in PE.” “Many students may consider themselves incapable of starting, when this is not true […] One of our roles has to be to break those beliefs and make them see that they can like the rest.” “What’s the point of achieving something if it can’t be shared with your peers? Challenges, when they are a team, make more sense and that is one of the main things I take away from the subject and methodology”* (GDE).


Another fundamental aspect that they have highlighted is obtaining resources about how to teach the classes and the fundamental role that PE should have in the schools:
*“I used to think that a teaching unit was simply a series of games and that’s it […] Now I’m seeing that a game or task in itself is not worth anything, it has to have a meaning to allow the student to think.” “There are many resources that I take with me from the subject, and that will undoubtedly be useful for me to develop in the classroom”* (DRG)*. “I believe that we have a very important challenge for the future, and that is to ensure that the school has a fundamental role in the schools.” “We can do many projects from the school that impact on the day-to-day life of the students […] Service-learning programs that promote health are important”* (GDE).


### 3.2. Attitudinal Style Transferability in School (257 text excerpts)

We can see how the students highlight the methodology received in the subject as fundamental to give rigor to PE and to what has to be taught in the subject:
*“The methodology received in the subject has helped me to realize that everything has to have a reason in our classes […] We could not expect all students to learn in the same way, and for that we as teachers have to give rigor to the subject.” “We must teach many types of content in class, but the key is in how we teach them”* (DRG)*. “What is the point of doing many tests and sports if the whole class does not enjoy and learn from them? I am clear that the importance lies in the success of the group and in shared responsibility”* (GDE).


Students also stressed the importance of relating PE to other subjects:
*“One of the fundamental things we are seeing in the subject is the importance of generating motivation in our students […]. That motivation is born when the task you propose is a challenge for the student, and that can be related to other subjects”* (DRG)*. “We have seen in the units how physical education can be related to other subjects, both at the level of content and at the level of motivation or social relationship […].” “The body is a fundamental agent of learning and must be present throughout the curriculum”* (GDE).


Finally, future PE teachers value receiving these methodologies at the university as something fundamental since they have a clear connection with the school and the extracurricular environment. In addition, they highlighted the possibility of hybridization with other approaches:
*“This is the first time we have received a methodology with such a practical component.” “David always shows us videos of students doing the things he proposes in class, and that always allows you to see the reality.” “We are a little tired of receiving subjects that talk about innovation and theories where we don’t see where or how they can be applied.” “This methodology in physical education has many possibilities in school and outside it, since it has the personality and emotions of the students very much in mind” (DRG). “I am left with many things, but above all with the way in which we have received the classes and the constant reflection of each activity. This reflection is fundamental to have as future teachers.” “This methodology is very interesting for the motivational part that it entails, and it can also be hybridized with other models, as we have seen in the didactic unit”* (GDE).


## 4. Discussion

The objectives of the study were twofold: (a) to delimit the characteristics and elements that compose Attitudinal Style as a pedagogical model; (b) to analyse the perception of future teachers about the utility and transferability of the model in their classes. The structural elements of the attitudinal style have been delimited, showing it as a model with a trajectory and great applicability in the area of PE. In this sense, the future teachers of PE have valued this model as ideal for the construction of their professional identity, with the social, motivational, and self-concept aspects being best valued for their transferability to the school.

The results have shown the importance that future teachers have given to the Attitudinal Style in order to position themselves as a way of understanding PE. It has helped them to reflect as professionals about the type of PE they consider necessary to teach in schools in the future. As indicated by Pérez-Pueyo and Hortiguela [83], we are in a moment in which it is necessary to be reflexive and critical about the postulates that revolve around PE, since there are too many passing fads that obviate the learning that is truly generated in the subject. We must bear in mind the teaching pillars that must structure the subject, and for this, it is necessary that from the initial teacher training we use the pedagogical models [13]. It is necessary to give coherence and consistency to the teaching processes of the PE, specifying the learning results to be achieved. This is the best way to be positively evaluated both on a curricular and social level [84].

Another one of the positive aspects that the participants have highlighted about the Attitudinal Style has been the social relationships generated throughout the course. The future teachers say they have seen how success in the achievement of tasks is more satisfactory when it is achieved in a group, totally changing the dynamics that are generated in class. Moreover, they have stressed that this fact allowed them to increase their self-concept and self-esteem. These results are in line with others [85], which reflect the social purposes that PE must have since motor experiences achieve their full meaning when they allow them to be shared with others. This positive social climate must be generated under three fundamental criteria: (a) student commitment to the task; (b) individual and group responsibility; (c) equitable distribution of work among group members [86]. If this is worked on in this way, the increase in the motivational climate of the group will be proportional to the emotional success obtained by each individual. 

Another key result of this study has been the great amount of resources that students have received thanks to the Attitudinal Style. In this way, they indicate the potential that the subject has both within and outside the curriculum. Bazana, McLaren, and Kabungaidze [87] indicate that one of the main purposes of the university should be the transferability of learning to society, and for this purpose the teaching should be structured based on the intentional development of skills of a reflective and professional nature. From PE this makes even more sense, since generating adherence to the practice of physical activity and sports means a great contribution to the health of an increasingly sedentary society [88]. In addition to these physiological benefits, the pedagogical character that characterizes PE allows for a high level of psychological and mental well-being, as long as it is approached from an angle that guarantees positive motor experiences in students [89].

Future PE teachers recognize that the Attitudinal Style goes far beyond performing a series of progressions of activities in class. It allows them to ask themselves why certain content resonates with students and what the learning objectives will be, and to work seamlessly with other subjects. This last aspect is fundamental, and more so in the primary stage where the motor aspect is ideal for optimal psycho-evolutionary development and the learning of contents associated with other subjects such as mathematics [90]. In this sense, future teachers have also highlighted the possibility of hybridizing the Attitudinal Style with other pedagogical models. This fact demonstrates their methodological capacity to interrelate key aspects in the classroom that have a direct impact on student learning. A pedagogical model should never have exclusive use by PE teachers, having demonstrated how the hybridization of the same allows it to adapt to the characteristics of the context, the content, and above all, the students [91]. It is fundamental that the initial training of PE teachers be structured based on three fundamental approaches: (a) rigorous theoretical questions linked to the discipline; (b) acquisition of methodological and evaluative resources applicable to the classroom; (c) critical reflection on the role that PE should have in schools [92]. Caldeborg, Maivorsdotter, and Öhman [93] indicate that there is still much disagreement about what the school should aim to achieve. This lack of agreement arises even amongst teachers who teach the subject, with the variables of motor commitment, type of content, and methodological orientation being the main sources of disagreement. In order to reach agreement among teachers, it is necessary to base discussions on scientific evidence about the learning outcomes of PE, which will allow us to advance rigorously every day in the classroom [94]. In this way, it will be possible to establish motor interventions in the classroom that are of a high pedagogical level, are coherent, and generate true and lasting learning.

## 5. Conclusions

In relation to the first objective of the study, the fundamental aspects that integrate the Attitudinal Style have been reflected, highlighting the groupings, the type of activities proposed by the teacher, the active role of the teacher, and the individual and group responsibility of the student as the most outstanding. Regarding the second objective, the future physical education teachers have valued very positively the Attitudinal Style, emphasizing its rigor when planning the subject and the great amount of methodological resources it offers. The main contribution of this study has been to propose the Attitudinal Style as a pedagogical model at an international level, providing results on the experiences of future EF teachers. However, it presents some limitations since it has only focused on teachers in training. Therefore, as a future line of research, it could be applied in the permanent training of teachers, checking what perceptions experienced PE teachers show. It could also be checked whether students who have received the Attitudinal Style show different perceptions depending on the content taught in the classroom. We consider that this article may be of interest to all those professionals teaching PE, especially those who train future teachers. If we consider the methodology as the main teaching tool, the Attitudinal Style becomes a fundamental pedagogical model to guide the educational processes in PE.

## Figures and Tables

**Table 1 ijerph-17-02816-t001:** Weekly information collection structure of the reflective group journals.

Weekly Information Collection Structure
1. What aspects have we dealt with this week in class (type of contents/tasks, organization of groupings, time management…)? 2. How has the methodology of the activities influenced the construction of my professional identity (reflection on the objectives set, structure of the sessions, adaptation of the tasks to the characteristics of the students…)? 3. What is the applicability of what we have seen in class to our professional future as teachers (transfer of the methodology to the school, usefulness of the tasks dealt with…)?

**Table 2 ijerph-17-02816-t002:** Basic script used for the final discussion group with the students.

The Script Used for the Final Discussion Group
Why do you think the subject you have studied is important? What methodological aspects do you highlight as fundamental? Why? What do you think the Attitudinal Style contributes in relation to other pedagogical models? What do you think the Attitudinal Style has contributed most to your understanding of PE? Do you think the Attitudinal Style is transferable to the school’s Physical Education? Why? What are the main benefits of this pedagogical model for children at school? After taking the course, have you changed your perception of the EF in any way? In what?

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
