# Peer review of "The Attitudinal Style as a Pedagogical Model in Physical Education: Analysis of Its Effects on Initial Teacher Training"

_ijerph, 2020, doi:10.3390/ijerph17082816_

Round 1
Reviewer 1 Report
This manuscript is generally well written, structured, and easy to follow. It manages to explain the major points well. The introduction could benefit from some more contemporary citations, otherwise it sets the scene well for the study. The instruments, measures and tables are appropriate.
Abstract: there is probably a typos by using the acronym FE, whereas it was previously stated as PE, standing for “physical education”. That inconsistent use of acronym is scattered throughout the entire manuscript (resulta, conclusions).
Introduction: I would personally try to tighten the introduction by condensing the two aims (paragraphs 1.1 and 1.2) in a more succinct way. There is not need to frame again each objective too much analytically.
Methods: - age of the subjects should be reported in units, i.e. years. A size sample determination should be conceived: samples appear small for current standards, in the face of replicability issues in Psychology. - instruments: please refer appropriately to the the construct validity of ‘robust’ theories behind the research questions investigated.
Results: a synoptic table would be likely of help.
Conclusions: Please may the Authors summarize in a few lines their achieved findings against the objectives of their research. Long-winded sentences do not lead the readership straight to the points. Finalize your work with a couple of informative concluding remarks.
Author Response
Dear editors and reviewers;
Thank you for your review and suggestions for improvement. We believe that after modifying the manuscript, taking into account the comments indicated, the article has undergone a notable improvement. Below we explain in detail each of the changes made.
Reviewer's comments
1.- This manuscript is generally well written, structured, and easy to follow. It manages to explain the major points well. The introduction could benefit from some more contemporary citations, otherwise it sets the scene well for the study. The instruments, measures and tables are appropriate.
Answer: We are very grateful for your words.
2.- Abstract: there is probably a typos by using the acronym FE, whereas it was previously stated as PE, standing for “physical education”. That inconsistent use of acronym is scattered throughout the entire manuscript (resulta, conclusions).
Answer: We have modified this error throughout the manuscript.
3.- Introduction: I would personally try to tighten the introduction by condensing the two aims (paragraphs 1.1 and 1.2) in a more succinct way. There is not need to frame again each objective too much analytically.
Answer: Sections 1.1 and 1.2 have been grouped together.
4.- Methods: - age of the subjects should be reported in units, i.e. years. A size sample determination should be conceived: samples appear small for current standards, in the face of replicability issues in Psychology. - instruments: please refer appropriately to the the construct validity of ‘robust’ theories behind the research questions investigated.
Answer: The average age of the subjects is given in years, with their respective standard deviation. This is a qualitative methodology, so the number of participants does not seek to be representative of the total (a clarification is made in this regard). With respect to the instruments, information has been added regarding their validity in relation to the objectives set.
5.- Results: a synoptic table would be likely of help.
Answer: A clarification has been made about the structure of the results and the way in which they are presented.
6.- Conclusions: Please may the Authors summarize in a few lines their achieved findings against the objectives of their research. Long-winded sentences do not lead the readership straight to the points. Finalize your work with a couple of informative concluding remarks.
Answer: The first lines of the conclusion set out the main results obtained from the two objectives of the study.
Reviewer 2 Report
Overall, I applaud the researchers for exploring the topic presented in this study. It is imperative that curriculum models are explored and taught during a student’s pre-service years. The ability to chose and adapt particular curriculum models that align with the student’s teaching philosophy and the context which they will be teaching in is key. I would like to pose a few questions to the authors:
- The majority of qualitative methodology requires you to discuss your positionality in regards to the study at hand. As a reader I am curious to know who the researchers are in relation to the subjects. Do the researchers hold an instructor of record position for some of the subjects and/or is the teacher of the Attitudinal Style course part of the research team. It is important to know because it will add some bias to the study that needs to be addressed. For example, do the subjects (students) receive a grade for their participation, since the teacher is reviewing the reflections weekly how does that affect the student’s willingness to be truthful in their responses?
- You state that 32 practical classes were used…it would be assumed that not all of these content choices would lend themselves so easily to the curriculum model you have chosen. Based off the data is there any data to show that students felt differently about the possibilities of implementing this model in varying content areas?
- Line 162: You state discussion groups were 12 students with high grades? Is this referring to the end of term grade and if so, did you select based on grade? If this is the case you might assume that students with lower grades might not have bought into the fact that this curricular choice is as valued. Would this have changed your responses/results of this study?
- Was triangulation met? If so, how?
Author Response
Dear editors and reviewers;
Thank you for your review and suggestions for improvement. We believe that after modifying the manuscript, taking into account the comments indicated, the article has undergone a notable improvement. Below we explain in detail each of the changes made.
Reviewer's comments
1.- Overall, I applaud the researchers for exploring the topic presented in this study. It is imperative that curriculum models are explored and taught during a student’s pre-service years. The ability to chose and adapt particular curriculum models that align with the student’s teaching philosophy and the context which they will be teaching in is key. I would like to pose a few questions to the authors:
Answer: We are very grateful for your words.
2.- The majority of qualitative methodology requires you to discuss your positionality in regards to the study at hand. As a reader I am curious to know who the researchers are in relation to the subjects. Do the researchers hold an instructor of record position for some of the subjects and/or is the teacher of the Attitudinal Style course part of the research team. It is important to know because it will add some bias to the study that needs to be addressed. For example, do the subjects (students) receive a grade for their participation, since the teacher is reviewing the reflections weekly how does that affect the student’s willingness to be truthful in their responses?
Answer: This information has been clarified in the sections on participants and on design and procedure.
3.- You state that 32 practical classes were used…it would be assumed that not all of these content choices would lend themselves so easily to the curriculum model you have chosen. Based off the data is there any data to show that students felt differently about the possibilities of implementing this model in varying content areas?
Answer: That's an interesting point. The Attitudinal Style addresses the same methodological guidelines, regardless of the content addressed. However, it is true that future physical education teachers may have different perceptions depending on the content taught. A future line of research on this subject is established within the conclusions.
4.- Line 162: You state discussion groups were 12 students with high grades? Is this referring to the end of term grade and if so, did you select based on grade? If this is the case you might assume that students with lower grades might not have bought into the fact that this curricular choice is as valued. Would this have changed your responses/results of this study?
Answer: This aspect has been clarified.
5.- Was triangulation met? If so, how?
Answer: This aspect has been clarified within the section of “Data analysis”.
All changes made have been incorporated in red within the manuscript.
Yours sincerely: The authors.